# Melanoidins from Chinese Distilled Spent Grain: Content, Preliminary Structure, Antioxidant, and ACE-Inhibitory Activities In Vitro

**DOI:** 10.3390/foods8100516

**Published:** 2019-10-18

**Authors:** Shiqi Yang, Wenlai Fan, Yan Xu

**Affiliations:** Key Laboratory of Industrial Biotechnology of Ministry of Education, Laboratory of Brewing Microbiology and Applied Enzymology, School of Biotechnology, Jiangnan University, 1800 Lihu Ave, Wuxi 214122, Jiangsu, Chinayxu@jiangnan.edu.cn (Y.X.)

**Keywords:** dried distilled spent grain (DDSG), melanoidins, content, structure, antioxidant activity, ACE-inhibitory activity

## Abstract

Distilled spent grain (DSG), as the major by-product of baijiu making, contains melanoidins generated via the Maillard reaction. In this study, four melanoidin fractions (RF1‒RF4) were isolated successively from dried DSG (DDSG) using sodium hydroxide solution and water as extractants, and the content, preliminary structure, and ACE-inhibitory activities in vitro of melanoidins were first investigated. The antioxidant activity was also evaluated. The results indicated that the total content of melanoidins was 268.60 mg/g DDSG dry weight (dw) using a model system of glucose and serine as standard, and RF4 showed the highest content of melanoidins (174.30 mg/g DDSG dw). Functional groups like C=O, N‒H, C‒N, O‒H, C‒H, C‒O, C-C, and ‒C‒CO‒C‒ were present in the structure of melanoidins from RF4, as determined by Fourier transform infrared (FT-IR) assay. The highest antioxidant activities, as assessed by 2,2′-azino-bis (3-ethylbenzothiazoline-6-sulfonic acid) (ABTS^•+^), ferric-reducing antioxidant power (FRAP), and 1,1-diphenyl-2-picrylhydrazyl radical (DPPH) assays, and the highest angiotensin-converting enzyme (ACE) inhibitory activity (95.92% at 2 mg RF4/mL) were also exhibited by RF4. The RF4 was further fractionated by ultrafiltration based on molecular weight (MW). The more than 100 kDa melanoidins (RF4-6) exhibited the highest yield and antioxidant activity. The 3‒10 kDa melanoidins (RF4-2) were more efficient in ACE-inhibitory activity. Our study could raise awareness of the DDSG as a value-added resource.

## 1. Introduction

Distilled spent grain (DSG), the sorghum residue obtained after solid state fermentation and distillation, is the biggest byproduct in baijiu (Chinese liquor) production. The output of DSG amounted to 30 million tons in 2018. DSG is rich in carbohydrates (5.71–11.34% *w*/*w*), protein (5.0–13.8% *w*/*w*), crude fiber (10.05–10.20% *w*/*w*), and crude lipids (1.31–3.24% *w*/*w*) [1]. DSG rots easily and causes serious environmental pollution due to the high water content (60~70%) and acidity. To date, DSG was mainly used for biogas production [2], livestock feeding [3], and single-cell protein production [4], etc.

In recent years, lots of studies have begun to focus on foods rich in bioactive compounds with the ability to promote benefits for human health [5]. Melanoidins from DSG, generated by the Maillard reaction of the baijiu-making process, are also bioactive compounds.

Melanoidins, a class of brown hydrophilic nitrogen-containing polymers, are formed in the last stage of the Maillard reaction, and they are widely found in coffee, cocoa, toast, dark beer, malt, honey, and other foods [6]. Apart from their role in the aroma and color as well as texture of thermally processed food, a lot of research on melanoidins has been carried out in recent years due to their potential nutritional, biological, and functional implications such as antioxidant [7], antihypertensive [8], prebiotic activity [9], and antimicrobial [10] properties.

The structure of melanoidins is complicated due to the strong influence of starting materials and reaction conditions [11]. The exact structure of melanoidins has remained unknown until now. There were three main proposals for the structure of melanoidins: (a) Melanoidins were polymers consisting of repeating units of furans and/or pyrroles [12]; (b) High molecular weight melanoidins were formed by the cross-linking of proteins with low molecular weight chromophores [13]; (c) The sugar degradation product, as the skeleton of melanoidins, could be linked to amino compounds to form melanoidins [14]. In model systems, it has been demonstrated that linear and branched melanoidin-like polymers in which bridging carbons link furan and pyrrole units exist [12] (Figure 1). Moreover, some compounds (mainly furans) accompanied by carbonyl compounds, pyrroles, pyrazines, pyridines, and some oxazoles were detected by thermal degradation in the glucose/glycine melanoidins [15] (Figure 2). As for real food, polysaccharides, proteins, and chlorogenic acids were reported to be involved in the formation of coffee melanoidins [16].

Some information about melanoidins from the spent grain of the brewing/liquor-making industry was available in brewers’ spent grain (BSG) and DSG. High molecular weight (MW) melanoidin (>10 kDa) extracts from black BSG was found to have relevance, with a high phenolic content, protein content, metal-chelating activity, and antioxidant activity [17]. The melanoidins from DSG have also been reported to possess antioxidant activity [18]. Up to now, few studies concerning the content, chemical composition, structure, and other biological activities of DSG melanoidins have been published. 

The purpose of this article was to study the content, preliminary structure, chemical properties, antioxidant, and angiotensin-converting enzyme (ACE) inhibitory activity of melanoidins from dried DSG (DDSG) in vitro. Our research may raise awareness of the DDSG as a value-added resource.

## 2. Materials and Methods

### 2.1. Materials

DSG samples used in this study were kindly provided by Golden Seed Distillery Co., Ltd. (Fuyang, China) and stored at −20 °C until use.

D-glucose, gallic acid, bovine serum albumin, methanol, and 1, 1-diphenyl-2-picrylhydrazyl radical (DPPH) were bought from Sigma-Aldrich (Shanghai, China). Total Antioxidant Capacity Assay Kits with a rapid 2,2′-azino-bis (3-ethylbenzothiazoline-6-sulfonic acid) (ABTS) and ferric-reducing antioxidant power (FRAP) were purchased from Beyotime Biotechnology Co., Ltd (Shanghai, China). ACE Kit-WST was obtained from Dojindo Inc. (Kumamoto, Japan). Other chemicals were analytical grade.

### 2.2. Melanoidin Extraction and Ultrafiltration

DSG was removed from −20 °C, thawed, and oven-dried at 70 °C for 24 h to a constant weight, and then the husk was removed. DDSG was crushed with a high-speed multipurpose grinder (RY-280A, Yongkang Ruiyi Electromechanical Co., Ltd, JinHua, China) and then filtered through a 0.25 mm mesh screen (Sinopharm Chemical Reagent Co., Ltd., Hefei, China).

The method of extracting melanoidins from DDSG was carried out according to previous studies [17,19] with some modifications. The extraction steps are schematically outlined in Figure 3. All centrifugation steps were carried out under the conditions of 2700× *g* at 10 °C for 20 min, and then all supernatants (RF1–RF4) were adjusted to pH 7.0 using 2 N NaOH or 2 N HCl as required. RF4 was further separated into different MW fractions by ultrafiltration using centrifugal tubes (Millipore Co., Ltd. Shanghai, China) equipped with 3, 10, 30, 50, and 100 kDa nominal MW cut-off membranes (MWCOs). All samples were centrifuged at 4000× *g* for 20 min. The retentate was washed with distilled water 3–5 times. Melanoidin fractions with different MWs were named as RF4-1 (<3 kDa), RF4-2 (3‒10 kDa), RF4-3 (10‒30 kDa), RF4-4 (30‒50 kDa), RF4-5 (50‒100 kDa), and RF4-6 (>100 kDa).

### 2.3. Preliminary Structural Identification

The preliminary structure of melanoidins from RF4 was identified by ultraviolet–visible absorption spectroscopy (UV-VIS) and Fourier transform infrared (FT-IR) spectra. UV-VIS was determined using a MAPADA P7 spectrophotometer (Mapada Instruments Co., Ltd., Shanghai, China) at room temperature with a scanning wavelength ranging from 200 to 800 nm. The FT-IR spectra were obtained by a FT-IR spectrometer (NEXUS470, NICOLET instruments co., Ltd., Madison, WI, USA) in the frequency range of 4000–600 cm^−1^. Lyophilizates of RF4 and six different MW fractions (RF4-1‒RF4-6) were made into potassium bromide flakes at similar weight concentrations before FT-IR analysis.

### 2.4. Color Analysis

The color analysis of melanoidin fractions was carried out by recording the absorbance using A380 spectrophotometer (AOYI instruments co., Ltd., Shanghai, China) with a 1-cm path length cell at 420 nm [20]. Distilled water was used as a blank.

### 2.5. Total Phenolic, Protein, and Carbohydrate Content

The total phenolic content (TPC) of melanoidin fractions (aqueous solutions of 1 mg/mL) was determined by the Folin-Ciocalteu procedures [21]. The TPC was calculated using a calibration curve with gallic acid (final assay concentration, 2–50 μg/mL) as standard. The results were expressed as mg gallic acid equivalents per g dry weight of DDSG (mg GAE/g DDSG dw).

The protein content of melanoidin fractions (aqueous solutions of 1 mg/mL) was measured by the Bradford method [22]. Bovine serum albumin was used as standard (0.01–0.5 mg/mL). The results were expressed as mg bovine serum albumin equivalents per g dry weight of DDSG (mg BSA/g DDSG dw).

The carbohydrate content of melanoidin fractions (aqueous solutions of 1 mg/mL) was estimated using the phenol‒sulfuric acid method [23]. A standard curve was plotted with glucose (0.01‒0.8 mg/mL) as standard. The results were expressed as mg glucose equivalents per g dry weight of DDSG (mg GE/g DDSG dw).

### 2.6. Total Antioxidant Capacity In Vitro

DPPH radical-scavenging capacity. The DPPH radical-scavenging capacity was measured in accordance with the described method [24]. Aliquots of 0.2 mL of sample solution (1 mg/mL) were added to 3.0 mL of DPPH solution (0.72 mM) prepared in methanol, then vortexed and allowed to stand in the dark for 30 min at room temperature, and the absorbance at 516 nm was read by a microplate reader (Cytation 3, BioTek Instruments, Inc., Winooski, VT, USA). The results were expressed as DPPH (%). The DPPH radical-scavenging rate (%) was calculated using Equation (1).
(1)$$\mathrm{DPPH}\;(\%)=\frac{A_{\mathrm{control}}-A_{\mathrm{sample}}}{A_{\mathrm{control}}}\times 100$$
A_sample_ is the absorbance of the sample solution, while A_control_ represents the absorbance of the blank in which 0.2 mL methanol replaced the sample solution.

Ferric-reducing antioxidant power (FRAP). The FRAP assay was achieved with the kit mentioned in the materials section. Briefly, aqueous solutions of FeSO4·7H2O (0.15‒1.5 mM) and a FRAP working solution were prepared prior to the assay. The FRAP working solution was obtained by mixing 150 μL of 2,4,6-tri(2-pyridyl)-s-triazine (TPTZ) diluent, 15 μL of TPTZ solution, and 15 μL of detection buffer sequentially for one measurement. When testing, an aliquot of 180 μL of FRAP working solution was added to each test well. Aliquots of 5 μL of sample solution (1 mg/mL) were added to sample wells, in blank wells was added 5 μL of distilled water instead, and 5 μL of FeSO4·7H2O was added to the wells for the standard curve. All of the wells were incubated at 37 °C for 3‒5 min. The absorbance was read at 593 nm. The results were expressed as mmol FeSO4 equivalents per g dry weight of DDSG (mmol FE/g DDSG dw).

ABTS^•+^ assay. The ABTS^•+^ assay was conducted using the kit mentioned in the materials section. The Trolox solution (0.15‒1.5 mM) used to make the standard curve and ABTS^•+^ working solution were prepared prior to testing. The ABTS^•+^ working solution consisted of 152 μL of assay buffer, 10 μL of ABTS^•+^ solution, and 8 μL of 1/1000 hydrogen peroxide solution for one measurement. An aliquot of 20 μL of peroxidase working solution was added to each test well. Then, 10 μL of the sample solution (1 mg/mL) was added to the sample well while 10 μL of distilled water was added to the blank well, and 10 μL of a different concentration of Trolox solution was added to the well of the standard curve. Finally, 170 μL of ABTS^•+^ working solution was added to each well and incubated at 37 °C for 3‒5 min at room temperature. The absorbance was recorded at 414 nm. The antioxidant capacity as measured with the ABTS^•+^ assay was expressed as mmol Trolox equivalents per g dry weight of DDSG (mmol TE/g DDSG dw).

### 2.7. ACE-Inhibitory Activity In Vitro

The determination of ACE-inhibitory activity was performed using a 96-well plate colorimetric detection system as previously described [25]. Each sample well contained the following solution: 20 µL of sample solution (2 mg/mL), 20 µL of substrate buffer, and 20 µL of enzyme working solution. Blank 1 consisted of 20 µL of distilled water, 20 µL of substrate buffer, and 20 µL of enzyme working solution. Blank 2 contained 40 µL of distilled water and 20 µL of substrate buffer. All of these wells were incubated at 37 °C for 1 h. Aliquots of 200 µL of the indicator working solution were added to each well and further incubated for 10 min. Then the absorbance was measured at 450 nm using a microplate reader. The formula for calculating the ACE inhibition rate was Equation (2).
(2)$$\mathrm{ACE}\;\mathrm{inhibition}\;\mathrm{rate}\;(\%)=\frac{A_{\mathrm{black1}}-A_{\mathrm{sample}}}{A_{\mathrm{blackl}}-A_{\mathrm{black2}}}\times 100$$
A_blank1_ is the absorbance without adding sample solution, A_blank2_ is the value without adding enzyme working solution, and A_sample_ represents the absorbance of the sample solution.

### 2.8. Calculation of Melanoidin Content

The method for calculating melanoidin content was performed according to [20] with some modification. The aliquot of 0.05 mol glucose and serine was fully dissolved with an appropriate amount of distilled water and freeze-dried to constant weight. The lyophilizate was placed in an oven at 90 °C for 1 h. Once reacted, brown solid was taken out and immediately cooled to room temperature then ground to fine powder, 5 g of which was dissolved in 200 mL distilled water. The solution was stirred for 12 h at 4 °C. After filtration, the content of melanoidins was calculated using the calibration curve obtained by using the filtrate as standard (0.1–5 mg/mL). The absorbance of the melanoidin standard was recorded at 420 nm.

### 2.9. Statistical Analysis

All tests in this study were measured at least in triplicate, and all statistical analyses were performed using IBM SPSS Statistics 22.0 (IBM, Armonk, NY, USA) software. Differences in mean were detected by one-way analysis of variance (ANOVA) after testing for normal distribution (Shapiro–Wilk test) and homogeneous variance (Levene’s test). Tukey’s test was then carried out, and values of *p* < 0.05 were considered to indicate significant difference.

## 3. Results and Discussion

### 3.1. The Content of Melanoidins in DDSG

Simulation of melanoidin standard is a method for estimating the amount of melanoidins in real foods. Model systems consisting of glucose and L-aspartic acid or glucose and glycine have served as standards of barley melanoidins [20,26]. Comparing the FT-IR spectra of the freeze-dried RF4 with model systems comprised of 20 amino acids and glucose respectively, the model system comprised of glucose and serine was selected as the standard of DDSG melanoidins.

Melanoidins can be rich in DDSG. The total amount of melanoidins in DDSG was 268.60 mg/g DDSG dw (Table 1), which was higher than the content in roasted barley malt (67.90 mg/g) [20], roasted coffee (72.00 mg/g), dry biscuit crusts (120.00 mg/g), and sliced bread crusts (180.00 mg/g) [27].

The content of melanoidins in four fractions (RF1–RF4) was significantly different (*p* < 0.05) (Table 1). RF1 represented the fraction obtained by an initial aqueous extraction of DDSG (Figure 3). RF4 was the fraction obtained using 110 mM NaOH extraction followed by isoelectric precipitation of proteins, and RF2 was the fraction obtained by extracting sediment with 1 N NaOH after 110 mM NaOH extraction, whereas RF3 was the fraction obtained by thoroughly aqueous washing of the precipitate extracted with 1 N NaOH. The amount of melanoidins in RF4 (174.30 mg/g DDSG dw) and RF2 (72.40 mg/g DDSG dw) were significantly higher than that of the other two fractions (Table 1), RF1 and RF3, suggesting that the fractions isolated by alkaline solution had a higher amount of melanoidins than fractions extracted by water.

### 3.2. The Preliminary Structure of Melanoidins

UV-VIS spectroscopy can provide certain information for the internal structure of organic compounds, and FT-IR assay is an ideal method for exploring protein–carbohydrate systems [28], both of which have been successfully implemented in the structural analysis of melanoidins [29,30].

RF4 was submitted to structural analysis due to having the highest amount of melanoidins. The UV-VIS spectrum (Figure 4) showed that the absorbance decreased with increasing wavelength, which was very similar to the descriptions previously reported [29,31,32]. The absorption peak at 280 nm indicated the possible presence of protein, and 420 nm was the characteristic absorption wavelength of melanoidins [33,34]. The absorption peaks were present both in ultraviolet and visible regions, suggesting that the structure of melanoidins contains a conjugated system.

The FT-IR spectrum (Figure 5) was similar to that of Chinese huangjiu (yellow wine) melanoidins [35]. The absorption peak at 1651.35 cm^‒1^ refers to C=O stretching vibration, N‒H bending vibration, and C‒N stretching vibration, which represents the amide I band of protein [36,37]. There were few peaks at 1540 cm^‒1^ (amide II) and 1300‒1200 cm^‒1^ (amide III). That was probably because the Maillard reaction changed the structure of protein. A wide and strong absorption band of O‒H stretching appeared around 3376.30 cm^‒1^, indicating that alcohol could be present in the structure of melanoidins. The characteristic absorption band near 2926.64 cm^‒1^ is assigned to the saturated C‒H stretching vibration, and the group is most likely to be –CH_2_‒ under this wave number. A weak absorption peak at 1402.67 cm^‒1^ could be generated by C‒N stretching vibration. The absorption peak at 1042.23 cm^‒1^ was generated by C‒O stretching vibration, C‒C stretching vibration, and C‒H bending vibration. The absorption peak around 565.13 cm^‒1^ could be generated by the in-plane bending vibration of the aliphatic ketone ‒C‒CO‒C‒ having a substituent at the α-position.

The FT-IR spectrum showed the possible presence of C=O, N‒H, C‒N, O‒H, C‒H, C‒O, C‒C, and ‒C‒CO‒C‒groups in the structure of DDSG melanoidins. To the best of our knowledge, the structure of DDSG melanoidins was first investigated in this study. Further structure analysis will be performed by thermal degradation, liquid chromatography-mass spectrometry (LC-MS), and nuclear magnetic resonance (NMR).

### 3.3. Color and Chemical Properties of Melanoidins

#### 3.3.1. Color Analysis

Brown is a typical external characteristic of melanoidins and might be caused by multiple chromophores [38]. The absorbance of four fractions (RF1‒RF4, aqueous solution of 1 mg/mL) was measured at 420 nm (Table 2). The RF4 exhibited the highest level of absorbance (0.507), followed by RF2 (0.188), RF3 (0.150), and RF1 (0.132).

#### 3.3.2. Chemical Properties Analysis

Phenols and protein as well as carbohydrate were considered to be the three major components of melanoidins [39,40,41]. The TPC and protein content of RF4 exhibited the highest levels (18.31 mg GAE/g DDSG dw and 88.98 mg BSA/g DDSG dw, respectively), followed by RF2 (8.73 mg GAE/g DDSG dw and 59.58 mg BSA/g DDSG dw) (Table 3). The fact that RF4 and RF2 possessed higher levels of phenols and protein is possibly due to the reason that alkaline extraction contributed to the release of phenolic compounds [42] and extraction of protein [43]. In addition, RF4 showed the highest carbohydrate content (164.30 mg GE/g DDSG dw). The higher levels of phenols and protein as well as carbohydrate exhibited by RF4 could be attributed to the presence of more melanoidins in RF4.

It is worth noting that there is no obvious absorption peak of phenol present in the FT-IR spectrum (Figure 5), and this could be due to the phenolic content of RF4 (18.31 mg GAE/g DDSG dw) being too low to be detected.

### 3.4. The Antioxidant and ACE-Inhibitory Activity of Melanoidins In Vitro

#### 3.4.1. Antioxidant Activity

Different methods used for antioxidant measurement could exhibit different experimental data, and two or more determined methods are needed herein to evaluate the antioxidant activity of melanoidins. Thus, ABTS^•+^, FRAP, and DPPH assays were chosen for determination. The results obtained by these three assays were similar in the RF1‒RF4 fractions, of which RF4 showed higher antioxidant capacity (ABTS^•+^, 1.03 mmol TE/g DDSG dw; FRAP, 0.33 mmol FE/g DDSG dw; DPPH, 20.57% at 1 mg RF4/mL) than the other three fractions (Table 4). The ability of RF4 to inhibit ABTS^•+^ oxidation was 38 times higher than the biscuits’ melanoidins [44] and 7 times higher than lightly roasted coffee melanoidins [7]. The antioxidant activity of RF4 as determined by the FRAP assay was higher than straw wine melanoidins (0.15‒0.26 mmol FE/g) [45]. However, the ability of RF4 to scavenge DPPH radicals (20.57%) was about one-third that of BSG melanoidins (59.50%) [19].

Melanoidins from DDSG could be used as a functional ingredient in the production of food due to its strong antioxidant activity.

#### 3.4.2. ACE-Inhibitory Activity

ACE (EC 3.4.15.1), a zinc-containing dipeptide carboxypeptidase, converts angiotensin I into angiotensin II in plasma, along with strengthening the contraction of the myocardium and increasing blood pressure. The inhibitory activity of ACE plays an important role in lowering blood pressure [46]. As shown in Table 5, RF4 exhibited the highest ACE-inhibitory activity, with a value of 95.92% at 2 mg RF4/mL. The ACE-inhibitory activity of RF4 was obviously stronger than that of melanoidins in coffee (lightly roasted coffee, 36.80%; medium-roasted coffee, 43.10%; dark-roasted coffee, 45.10%) [8], vinegar (15.75‒43.68%) [47], and black brewer’s spent grain (75.18%) at 2 mg/mL [17]. It has been reported that the ACE-inhibitory activity of melanoidins is related to antioxidant activity [48]. It would be interesting to study whether the ACE-inhibitory activity of melanoidins from DDSG is related to its high antioxidant activity under ABTS^•+^ and FRAP assays.

### 3.5. Preliminary Structure and Properties of Different MW Melanoidin Fractions from RF4

RF4 had the deepest color, the highest content, the strongest antioxidant capacity and ACE-inhibitory ability of melanoidins compared to the other three fractions. The different MW subfractions of RF4, RF4-1 (<3 kDa), RF4-2 (3‒10 kDa), RF4-3 (10‒30 kDa), RF4-4 (30‒50 kDa), RF4-5 (50‒100 kDa), and RF4-6 (>100 kDa), were obtained by ultrafiltration for further analysis of the structure and properties.

#### 3.5.1. Yield of RF4 Subfractions

The yields of RF4 subfractions are presented in Table 6, and it is obviously observed that RF4-6 (>100 kDa) had the highest yield (12.76% of the DDSG dw), suggesting that most of the melanoidins in RF4 from DDSG were >100 kDa compounds. The more than 100 kDa melanoidins have also been reported in real foods like coffee [49], roasted malt [50], and bread crusts [6]. Furthermore, the formation of high MW melanoidins (MW > 10 kDa) was related to an intense heating temperature and long reaction time [51]. Therefore, the formation of melanoidins in DDSG could be derived from the higher temperature daqu, the higher temperature stacking of fermenting grains, and distillation processes.

#### 3.5.2. Color and Chemical Properties of RF4

The RF4 subfractions were characterized in terms of their color, phenolic content, protein content, and carbohydrate content (Table 7). RF4-6 exhibited the darkest color (0.633, 1 mg RF4-6/mL), the highest content of phenols (4.66 mg GAE/g DDSG dw), carbohydrate (61.48 mg GE/g DDSG dw), and protein (34.17 mg BSA/g DDSG dw), followed by RF4-5 and RF4-4. RF4-1 had the lightest color (0.128), lowest phenols (1.41 mg GAE/g DDSG dw), carbohydrate (5.41 mg GE/g DDSG dw), and protein content (2.40 mg BSA/g DDSG dw). The higher absorbance exhibited by higher MW fractions, except for RF4-2, which is indicative of an increase in color contribution as MW increased. This phenomenon was in accordance with previous work [30]. The total phenols and protein content, as well as carbohydrate content, were more concentrated in RF4-6, which had the highest yield of melanoidins, further proving the inference mentioned above, that these three compositions could be involved in the melanoidin structure.

#### 3.5.3. Preliminary Structure of RF4 Subfractions

At the wavelength of 200‒800 nm, the UV-VIS spectra of RF4 subfractions showed similar shapes but different absorbance (Figure 6), suggesting that different MW melanoidin fractions had similar characteristics in terms of structure but different amounts of chromophores.

The FT-IR assay of different MW melanoidin fractions showed that the structure of these MW fractions could be similar, except for RF4-6 (Figure 7). It was speculated that different MW melanoidin fractions contained the same functional groups (C=O, N‒H, C‒N, O‒H, C‒H, C‒O, C‒C, and ‒C‒CO‒C‒), except for RF4-6. RF4-6 had an obvious absorption peak at 1519 cm^‒1^, which could be caused by the presence of a benzene ring according to previous research on the structure of melanoidins [16].

#### 3.5.4. Physiological Activity of RF4 Subfractions In Vitro

The antioxidant capacity of different MW fractions was investigated under ABTS^•+^, FRAP (Figure 8a), and DPPH assays (Figure 8b). Except for the similar antioxidant capacity of RF4-2 and RF4-3, the other MW fractions have statistically significant differences (*p* < 0.05). On the whole, the antioxidant capacity of RF4 subfractions increased with the increase in MW. RF4-6 showed the highest antioxidant capacity, while RF4-1 had the lowest. The ability of RF4-6 to scavenge ABTS^•+^ (0.23 mmol TE/g DDSG dw) was 1.8 times higher, FRAP (0.13 mmol FE/g DDSG dw) was 2.5 times higher, and the DPPH radical-scavenging rate (39.52% at 1 mg RF4-6/mL) was 26 times higher compared with RF4-1. RF4-2 with lower MW had higher antioxidant activity than RF4-3, possibly because RF4-2 contained more phenols. The phenols of melanoidins can show high antioxidant activity by binding to melanoidin skeletons with non-covalent bonds [33]. In the case of the DPPH radical-scavenging rate, the fractions with MW < 30 kDa had little scavenging activity, and DPPH radical-scavenging rate of MW > 100 kDa fractions could reach about 40%.

The ACE-inhibition rate of different MW fractions ranged from 54.64% to 92.74% (Figure 9). For fractions with MW > 3 kDa, the ACE-inhibitory activity gradually decreased with the increase of MW, which was contrary to the phenomenon previously observed for antioxidant activity. The fractions with MW 3‒10 kDa (RF4-2) had the strongest ACE-inhibitory activity, at 92.74%.

## 4. Conclusions

In this work, preliminary explorations of the content, structure, antioxidant capacity, and ACE-inhibitory activity of DDSG melanoidins were reported. The determination of content was achieved through a model system of serine and glucose. The total content of melanoidins was 268.60 mg/g DDSG dw, and melanoidins of DDSG mainly comprised >100 kDa compounds (12.76% of the DDSG dw). Functional groups such as C=O, N‒H, C‒N, O‒H, C‒H, C‒O, C‒C, and ‒C‒CO‒C‒ were present in the structure of melanoidins. In addition, DDSG melanoidins were proven to possess strong FRAP, ABTS^•+^ scavenging capacity, and ACE-inhibitory activity. The large availability of DSG and the health benefits of melanoidins suggested herein could provide new possibilities for its high value-added utilization. Future work on the main production stage of DSG melanoidins and changes in the baijiu-making process is worth carrying out.

## Figures and Tables

**Figure 1 foods-08-00516-f001:**
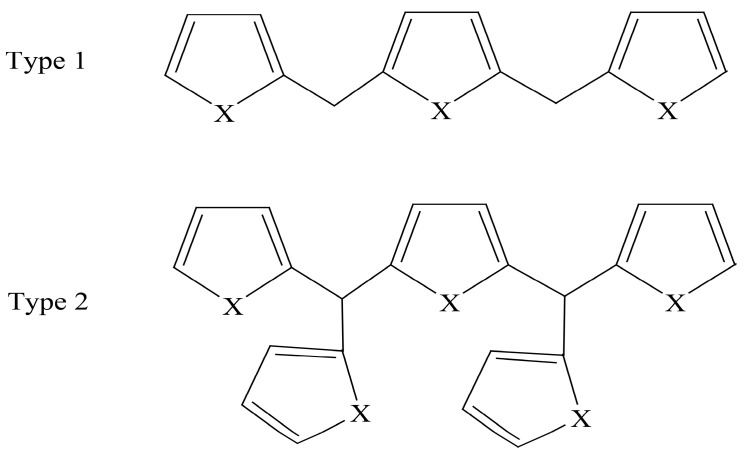
Proposed structures for melanoidin polymers, X = NR or O [12].

**Figure 2 foods-08-00516-f002:**
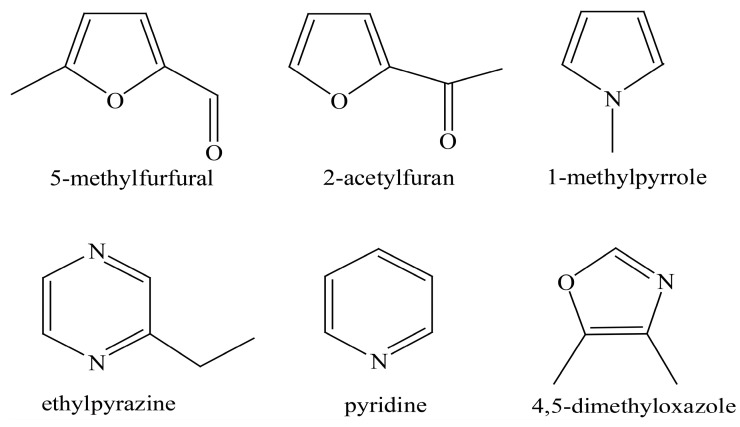
Structures of some representative compounds identified in the glucose/glycine melanoidins [15].

**Figure 3 foods-08-00516-f003:**
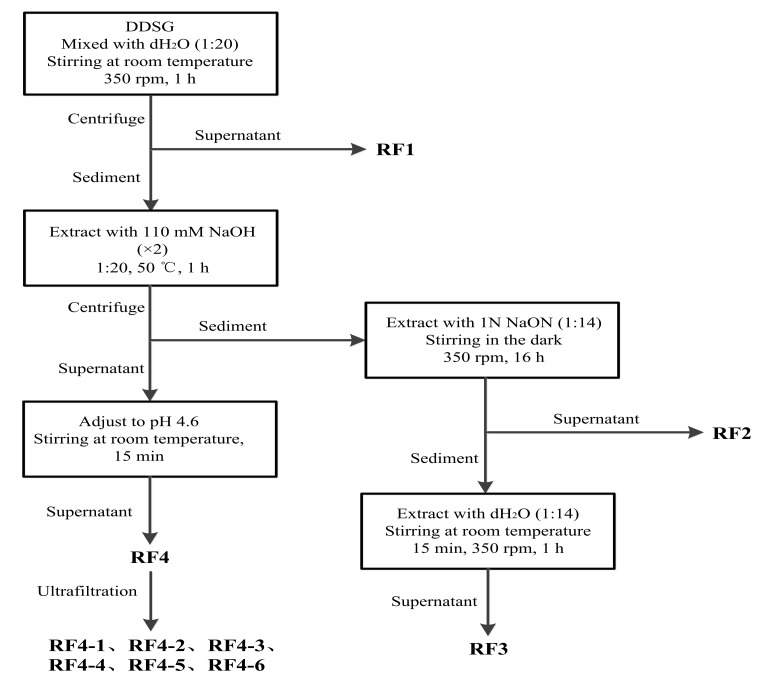
Flowchart for extraction melanoidins from dried distilled spent grain (DDSG). Four supernatants (RF1–RF4) containing melanoidins were obtained, and six different molecular weight (MW) fractions named RF4-1 (<3 kDa), RF4-2 (3‒10 kDa), RF4-3 (10‒30 kDa), RF4-4 (30‒50 kDa), RF4-5 (50‒100 kDa), and RF4-6 (>100 kDa) were obtained by ultrafiltration.

**Figure 4 foods-08-00516-f004:**
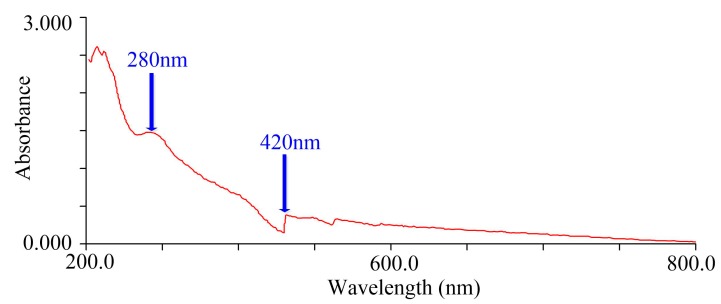
Ultraviolet-visible absorption spectroscopy (UV-VIS) spectrum of RF4. The aqueous solutions of RF4 was 1 mg/mL.

**Figure 5 foods-08-00516-f005:**
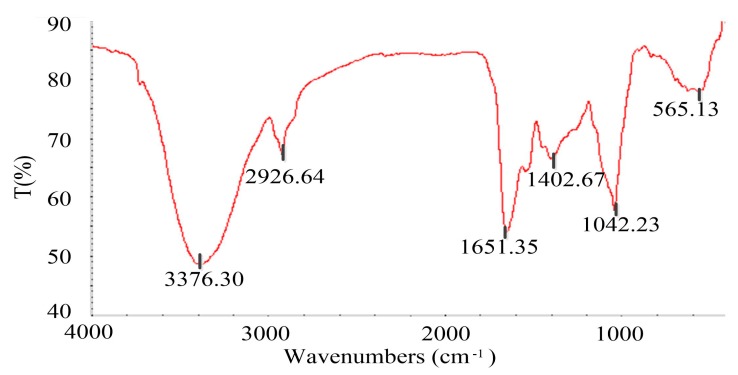
Fourier transform infrared (FT-IR) spectrum of RF4. The wavenumbers corresponding to the important absorption peaks were marked in the figure.

**Figure 6 foods-08-00516-f006:**
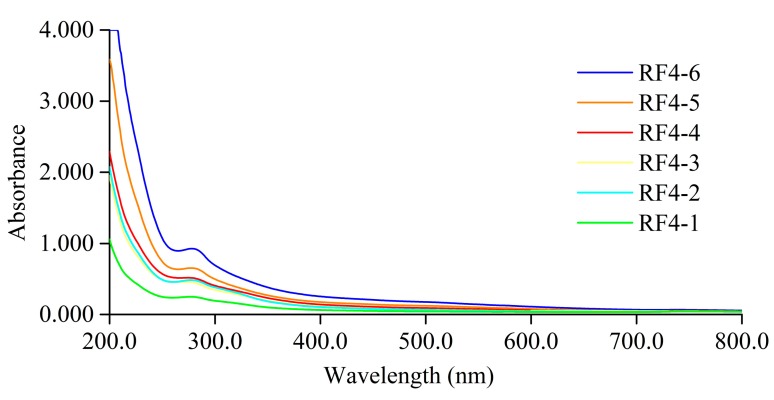
UV-VIS spectrum of different MW fractions obtained from RF4 by ultrafiltration. The aqueous solutions of fractions were 0.2 mg/mL.

**Figure 7 foods-08-00516-f007:**
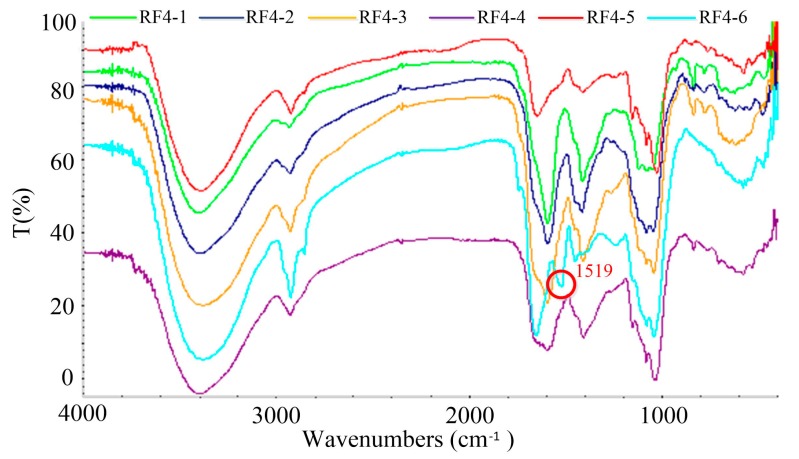
FT-IR spectrum of different MW fractions obtained from RF4 by ultrafiltration.

**Figure 8 foods-08-00516-f008:**
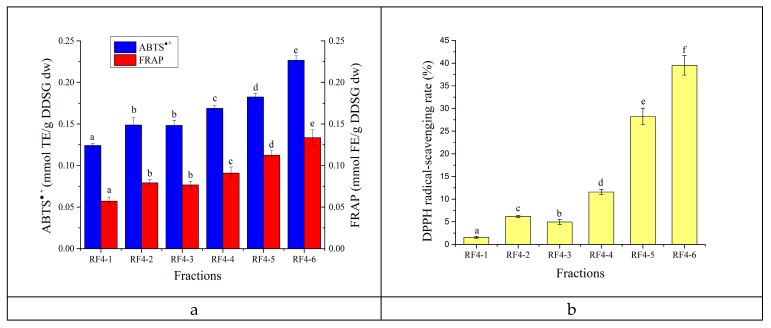
The antioxidant capacity of different MW fractions assessed by ABTS^•+^ (**a**), FRAP (**a**), and DPPH assay (**b**). The aqueous solutions of fractions were 1 mg/mL. Differences in mean were detected by ANOVA after conducting a Shapiro-Wilk test and Levene’s test. Different letters in the same assay represent significant differences (*p* < 0.05).

**Figure 9 foods-08-00516-f009:**
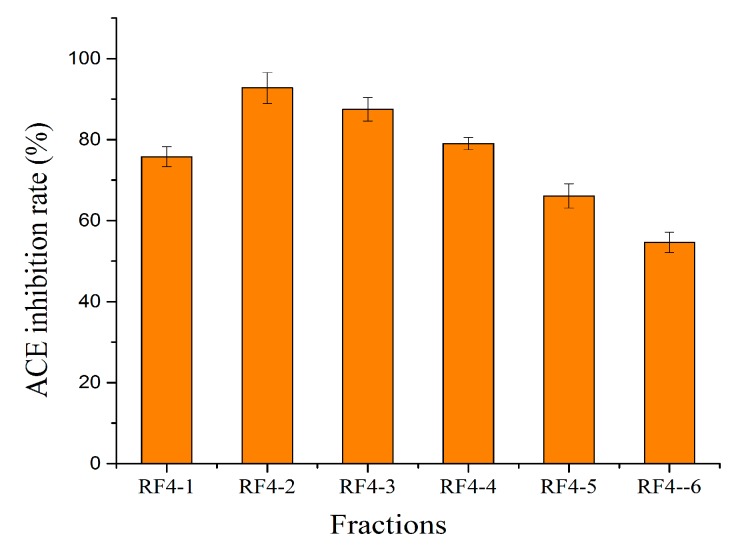
The ACE-inhibitory activity assay of different MW fractions. The aqueous solutions of fractions were 2 mg/mL.

**Table 1 foods-08-00516-t001:** Content of melanoidins in four fractions obtained by alkaline solution (RF2 and RF4) and distilled water (RF1 and RF3) extraction of dried distilled spent grain (DDSG).

Fractions	Content (mg/g DDSG Dry Weight (dw)) ^1^
RF1	9.69 ± 1.85^d^
RF2	72.40 ± 3.67^b^
RF3	12.20 ± 1.59^c^
RF4	174.30 ± 5.80^a^
total	268.60

^1^ The results are shown as mean ± standard deviation (SD) (*n* = 3). Differences in mean were detected by one-way analysis of variance (ANOVA) after testing for normal distribution (Shapiro–Wilk test) and homogeneous variance (Levene’s test). Values in the same column with different letters are significantly different (*p* < 0.05).

**Table 2 foods-08-00516-t002:** Absorbance of RF1‒RF4 at 420 nm ^1^.

Fractions	RF1	RF2	RF3	RF4
Absorbance	0.132 ± 0.02	0.188 ± 0.06	0.150 ± 0.05	0.507 ± 0.08

^1^ The absorbances of four fractions were recorded at aqueous solutions of 1 mg/mL, and the results are shown as mean ± SD (*n* = 3).

**Table 3 foods-08-00516-t003:** Content of total phenols, protein, and carbohydrate in four fractions ^1^.

Fractions	Total Phenols (mg GAE/g DDSG dw)	Protein (mg BSA/g DDSG dw)	Carbohydrate (mg GE/g DDSG dw)
RF1	5.23 ± 0.07 ^c^	14.15 ± 0.64 ^d^	131.90 ± 4.62 ^b^
RF2	8.73 ± 0.10 ^b^	59.58 ± 2.41 ^b^	61.84 ± 2.02 ^c^
RF3	5.56 ± 0.13 ^c^	50.01 ± 4.99 ^c^	32.93 ± 1.04 ^d^
RF4	18.31 ± 1.23 ^a^	88.98 ± 3.89 ^a^	164.30 ± 3.27 ^a^

^1^ The determinations were performed at the aqueous solutions of 1 mg/mL. The results are shown as mean ± SD (*n* = 3). Differences in mean were detected by ANOVA after conducting a Shapiro–Wilk test and Levene’s test. Values in the same column with different letters are significantly different (*p* < 0.05).

**Table 4 foods-08-00516-t004:** Antioxidant activity of RF1‒RF4 ^1^.

Fractions	ABTS^•+^ (mmol TE/g DDSG dw)	FRAP (mmol FE/g DDSG dw)	DPPH (%)
RF1	0.88 ± 0.01 ^b^	0.04 ± 0.02 ^c^	4.90 ± 0.02 ^c^
RF2	0.89 ± 0.02 ^b^	0.11 ± 0.04 ^b^	10.63 ± 0.39 ^b^
RF3	0.87 ± 0.01 ^b^	0.04 ± 0.03 ^c^	5.88 ± 0.26 ^c^
RF4	1.03 ± 0.03 ^a^	0.33 ± 0.03 ^a^	20.57 ± 0.92 ^a^

^1^ The aqueous fractions used in ABTS^•+^, FRAP, and DPPH assays were 1 mg/mL. The results are shown as mean ± SD (*n* = 3). Differences in mean were detected by ANOVA after conducting a Shapiro–Wilk test and Levene’s test. Values in the same column with different letters are significantly different (*p* < 0.05).

**Table 5 foods-08-00516-t005:** Angiotensin-converting enzyme (ACE)-inhibitory activity of RF1‒RF4 ^1^.

Fractions	RF1	RF2	RF3	RF4
ACE inhibition rate (%)	86.43 ± 3.42	63.83 ± 5.18	88.28 ± 3.59	95.92 ± 4.86

^1^ The aqueous fractions used in the assay of ACE inhibition rate were 2 mg/mL. The results are shown as mean ± SD (*n* = 3).

**Table 6 foods-08-00516-t006:** Yields of melanoidin fractions with different molecular weights (MWs).^1^

Fractions	Yield (%)
RF4-1 ^2^	0.64
RF4-2	2.46
RF4-3	0.06
RF4-4	0.37
RF4-5	0.94
RF4-6	12.76

^1^ The yields were calculated on a DDSG dw basis. ^2^ RF4-1: <3 kDa, RF4-2: 3‒10 kDa, RF4-3: 10‒30 kDa, RF4-4: 30‒50 kDa, RF4-5: 50‒100 kDa, RF4-6: >100 kDa.

**Table 7 foods-08-00516-t007:** Absorbance, total phenolic, carbohydrate, and protein content of different MW fractions ^1^.

Fractions	Absorbance	Total Phenols (mg GAE/g DDSG dw)	Carbohydrate (mg GE/g DDSG dw)	Protein (mg BSA/g DDSG dw)
RF4-1 ^2^	0.128 ± 0.01 ^d^	1.41 ± 0.06 ^e^	5.41 ± 0.71 ^f^	2.40 ± 0.38 ^e^
RF4-2	0.144 ± 0.01 ^d^	1.93 ± 0.38 ^d^	11.62 ± 0.65 ^e^	4.09 ± 0.52 ^de^
RF4-3	0.135 ± 0.02 ^d^	1.76 ± 0.13 ^de^	15.44 ± 0.87 ^d^	7.41 ± 0.98 ^d^
RF4-4	0.302 ± 0.01 ^c^	2.50 ± 0.38 ^c^	28.70 ± 2.86 ^c^	16.12 ± 1.06 ^c^
RF4-5	0.359 ± 0.03 ^b^	3.03 ± 0.43 ^b^	40.13 ± 3.05 ^b^	23.85 ± 0.82 ^b^
RF4-6	0.633 ± 0.05 ^a^	4.66 ± 0.13 ^a^	61.48 ± 5.15 ^a^	34.17 ± 3.31 ^a^

^1^ Determinations were carried out on different aqueous fractions of 1 mg/mL, and the results are shown as mean ± SD (*n* = 3). Differences in mean were detected by ANOVA after conducting a Shapiro-Wilk test and Levene’s test. Values in the same column with different letters are significantly different (*p* < 0.05). ^2^ RF4-1: <3 kDa, RF4-2: 3‒10 kDa, RF4-3: 10‒30 kDa, RF4-4: 30‒50 kDa, RF4-5: 50‒100 kDa, RF4-6: >100 kDa.

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
