# Peer review of "Melanoidins from Chinese Distilled Spent Grain: Content, Preliminary Structure, Antioxidant, and ACE-Inhibitory Activities In Vitro"

_foods, 2019, doi:10.3390/foods8100516_

Round 1

Reviewer 1 Report

1) What is the novelty of the study? I think that the aim of the work should be improved.

2) Tables 4 and 7 - ANOVA should be done such as in Table 3.

Reviewer 2 Report

The manuscript focuses mainly on an investigation into antioxidant activity of four melanoidins-containing fractions obtained from distilled spent grain. Some enzyme inhibition studies were also carried out complementary to the antioxidant activity of the fractions tested. Also, an attempts for structural elucidaton of the melanoidins was made using few spectroscopic methods.

The overall design of the manuscript is good, however there are serious issues which have to be addressed.

The introduction section lacks a figure depicting the structural features relevant to melanoidins. A wider explanation of Millard reaction in relation to melanoidins structures should be provided in the introduction section. Also, a relevant figure is highly recommended. The structure of melanoidins section is the weakest part of the manuscript. Authors provide only a general information on characteristic UV-Vis and IR bands, which is highly insufficient. In this context, the UV-Vis absorption spectra presented in the manuscript are particularly inconclusive. In order to characterize the structures of melanoidins present in the fractions tested a more sophisticated techniques such as HPLC-MS have to be employed. The LC-MS profile of melanoidins detected should be at least referenced to online libraries of known compounds. The conclusion section should be improved accordingly.

Reviewer 3 Report

All comments are in manuscript.

Round 2

Reviewer 2 Report

The current version of the manuscript is an improvement compared to the former one. Most issues are properly addressed.